## REVIEW ARTICLE

# Gliomas: a reflection of temporal gliogenic principles

Caitlin Sojka [1] & Steven A. Sloan [1,2] ✉

The hijacking of early developmental programs is a canonical feature of gliomas where neoplastic cells resemble neurodevelopmental lineages and possess mechanisms of stem cell resilience. Given these parallels, uncovering how and when in developmental time glioma-genesis intersects with normal trajectories can greatly inform our understanding of tumor biology. Here, we review how elapsing time impacts the developmental principles of astrocyte (AS) and oligodendrocyte (OL) lineages, and how these same temporal programs are replicated, distorted, or circumvented in pathological settings such as gliomas. Additionally, we discuss how normal gliogenic processes can inform our understanding of the temporal progression of gliomagenesis, including when in developmental time gliomas originate, thrive, and can be pushed towards upon therapeutic coercion.

Normal glial development is guided by a series of tightly controlled and temporally regulated lineage-determining events. Gliogenic malignancies represent a perturbation in this process, where neurodevelopmental programs are hijacked under severe genetic and environmental settings leading to aberrant cell fate decisions and pathogenic consequences. In this review, we highlight where normal and oncogenic glial differentiation paths diverge and how this information can uncover new facets of tumor biology. First, we summarize the current understanding of the timeline and instructive cues that define normal astro- and oligo-genesis. We then provide an overview of how molecular regulators of glial development and glial hierarchical organization are mimicked across glioma subtypes. Finally, we consider the ways in which normal glial differentiation can inform our understanding of how glioma cells move across developmental trajectories, including discussion on when in developmental time gliomas begin, progress to, and are capable of differentiating towards.

**Molecular regulators of astrocyte and oligodendrocyte lineage commitment**. Neuroecto-dermal development involves multipotent neural stem cells (NSC) called radial glia (RG) that give rise to three key cell populations—neurons, astrocytes (AS), and oligodendrocytes (OL). RG first undergo symmetrical divisions during early gestation to expand their pool before dividing asymmetrically towards neurogenic fates during mid-gestation, followed by gliogenic fates (astrocytes and oligodendrocytes) at later gestational and early postnatal stages[1,2]. This shift from neurogenesis to gliogenesis, termed the gliogenic switch, occurs around 16 gestational weeks (GW) in humans and is mediated by both intrinsic and extrinsic factors that synergize to suppress neurogenesis, release molecular brakes impeding gliogenesis, and actively promote gliogenic commitment[3,4]. For the purpose of this review, references to glia specifically pertain to astrocyte and oligodendrocyte macroglial populations.

During the neurogenic phase of development, premature astrogenesis is primarily prevented through inhibition of the JAK/STAT pathway and more specifically, STAT3-mediated transcription of astrocyte genes, including GFAP and S100B[4–7]. Pro-neuronal transcription factors (TF), such as NGN1[8], and the neurotrophin BDNF[9], both inhibit STAT3-mediated

---

[1] Department of Human Genetics, Emory University School of Medicine, Atlanta, GA, USA. [2] Emory Center for Neurodegenerative Disease, Emory University School of Medicine, Atlanta, GA, USA. ✉email: sasloan@emory.edu

astrogenesis while simultaneously promoting neurogenic pathways, like MEK-ERK signaling[10,11]. These mechanisms ensure a robust population of early immature neurons prior to the emergence of astrocytes and oligodendrocytes.

A key event that drives the shift towards gliogenesis is the remodeling of regulatory genomic regions into favorable states that promote the transcription of gliogenic genes. During astrogenesis, this occurs through synergistic activation of the JAK/STAT, BMP, and Notch signaling pathways, which modulate the landscape of DNA methylation, histone methylation, and acetylation[4,12–15]. The p300/CBP complex is an important component of the JAK/STAT pathway and has intrinsic acetyltransferase activity, including helping to induce H3K9 and H3K14 acetylation at the STAT3 binding site of the GFAP promoter[7]. Around the time of the gliogenic switch, Polycomb group (PcG) proteins silence NGN1 activity, inducing the release of p300/CBP, which forms a co-activator complex with STAT3 at the promoter of astrocyte genes to activate expression[12,13]. Additionally, the binding of astrocytic TFs, such as NFIA, has been shown to displace DNMT1 from astrocyte-specific promoters, helping to facilitate an active gliogenic transcriptional state[16–19].

Transcription factors are powerful molecular regulators that initiate changes in cell state, differentiation, and maturation. Advancements in high-throughput sequencing coupled with new and robust methods for studying glia—such as sophisticated 2D and 3D model systems, improved glial purification methods, and more specific genetic targeting of glia[20,21]—helped identify several TFs that contribute to gliogenesis. Two of the first TFs that were identified as key players in the induction of astrogenesis include NFIA and SOX9. Overexpression of NFIA is sufficient to induce astrocyte formation[22,23] and also drives HES5 expression, a Notch pathway effector required for the inhibition of neurogenesis[24]. Similarly, reduced Sox9 expression results in prolonged neurogenesis and delayed gliogenesis in vitro[25]. Kang and colleagues later discovered that Sox9 not only induces NFIA expression but identified that the two TFs form a complex to facilitate transcription of astrocyte genes[26]. Two additional Sox9 binding partners, NFIB and Zbtb20, also collectively induce cortical astrocyte differentiation in mice[27,28]. Several studies have subsequently identified key regulators of the SOX9-NFIA complex, including TFs PITX1, which promotes SOX9 expression[29], and Brn2, which plays a key role in SOX9-induction of NFIA[30]. Together, this illustrates a complex network of TF activation that is required to promote the switch from neurogenesis to astrogenesis (Fig. 1). Several studies have also investigated the role of TFs at later stages of astrocyte maturation, although this developmental window remains comparatively more elusive. Work by Lattke and colleagues suggested Rorb, Dbx2, Lhx2, and Fezf2 are potential regulators of astrocyte maturation in the developing mouse cortex[31]. However, substantial changes in maturation were more apparent when all four TFs were simultaneously overexpressed. Most likely, these TFs, as well as yet-to-be-identified candidates, act synergistically and/or in physical complexes to promote maturation.

The gliogenic switch is a shift not only from neurogenic to astrogenic fates but also towards oligodendrocyte lineages. Several TFs are implicated in early oligodendrocyte precursor cell (OPC) development and maintenance (Fig. 1). The TF Olig2, for example, activates additional OL-lineage TFs, including Sox10[32], which acts in combination with Sox9 to promote OPC maintenance and proliferation[33]. Co-deletion of Sox9 and Sox10 reduces the density of Olig2-positive OPCs within the developing spinal cord, and the remaining Olig2-positive OPCs are deficient in Pdgfrα, a signaling pathway that promotes OPC survival and proliferation[33,34]. In return, Sox10 helps maintain Olig2 expression in a positive feedback loop, together supporting the

maintenance of a robust OPC population[35]. Additionally, OPCs express several TFs, including Sox5, Sox6, Hes5, Id2, and Id4, which prevent OPC differentiation and maturation by inhibiting Olig1/2, Sox10, and downstream transcription of key maturation genes[36–38] (Fig. 1).

**Extrinsic regulators of gliogenesis**. In addition to intrinsic regulators of cell fate, multiple extrinsic cues are also important for promoting gliogenic commitment and downstream glial development[39]. Some of the most well-documented intrinsic factors in astrogenesis include a trio of IL-6 cytokines—cardiotrophin-1 (CT-1), leukemia inhibitory factor (LIF), and ciliary neurotrophic factor (CNTF)—that promote astrocyte formation through JAK/STAT activation[40–42]. Newborn neurons also secrete ligands Jagged 1 and Delta-like 1, which contribute to the gliogenic switch through activation of Notch signaling[18]. Multiple cytokines, including BMP2, BMP4, and TGF-B1, have also been implicated in astrogenesis by promoting the formation of a Smad:p300/CBP:STAT complex that facilitates the transcription of astrocyte genes[43–47]. FGF2[48,49] and retinoic acid (RA)[50] may act more broadly to promote astrogenesis by facilitating shifts in chromatin state to elicit transcription of astrocyte genes. Additionally, synergistic activity of the ligands TGFβ2, NLGN1, TSLP, DKK1, and BMP4 act upstream of mTORC1 to promote astrocyte development, suggesting that much like TFs, astrogenesis is orchestrated by a concert of extrinsic cues[51].

Extrinsic cues also play a substantial role in OL development, including PDGF-α, FGF-2, and IGF-1 signaling. PDGF-α is secreted by both neurons and astrocytes and helps maintain the OPC population by promoting proliferation and preventing precocious differentiation[33,34,52,53]. When the PDGF mitogen binds to and activates PDGF receptors, it triggers a reorganization of the actin filament structure, stimulating changes in cell growth and motility, a cascade that when hyperactivated can serve as an oncogenic program[54]. The mitogen FGF-2 helps to maintain the expression of PDGFRα and blocks oligodendrocyte differentiation by downregulating major myelin proteins[55–57]. FGF-2 and PDGFα, in combination with IGF-1, also work synergistically to promote OPC DNA synthesis and proliferation to ensure continual replenishing of OPC populations[58,59].

**Glial maturation**. After populating the CNS, astrocytes undergo a profound maturation process, evidenced by changes in gene expression, morphology, and function. In the first month of rodent postnatal development astrocyte appearance shifts from cells with simple filopodial processes that overlap with neighboring astrocytes to dense elaborate branching where cells occupy spatially segregated non-overlapping domains, a process referred to as tiling[60–63]. Likewise, recent studies using mouse models[31] and primary human fetal tissue samples[64,65] have identified thousands of differentially expressed genes (DEGs) between prenatal astrocyte precursor cells and postnatal astrocytes, highlighting differences in physiology and function between these two maturation states. For instance, immature astrocytes express high levels of proliferation genes TOP2A and MKI67, consistent with a developmental window when these cells are populating the CNS. During this time, immature astrocytes promote neuron migration and axon pathfinding[66,67], as well as guide synapse formation[68–70] and elimination[71,72]. While immature astrocytes help guide CNS construction, mature astrocyte functions shift towards supporting a homeostatic state. This is evidenced by the upregulation of gap junction (GJA1 and GJB6) and water channel (AQP4) genes in mature astrocytes, which are important for mediating neuronal signaling and meeting the fluctuating metabolic demands of the CNS[73–75].

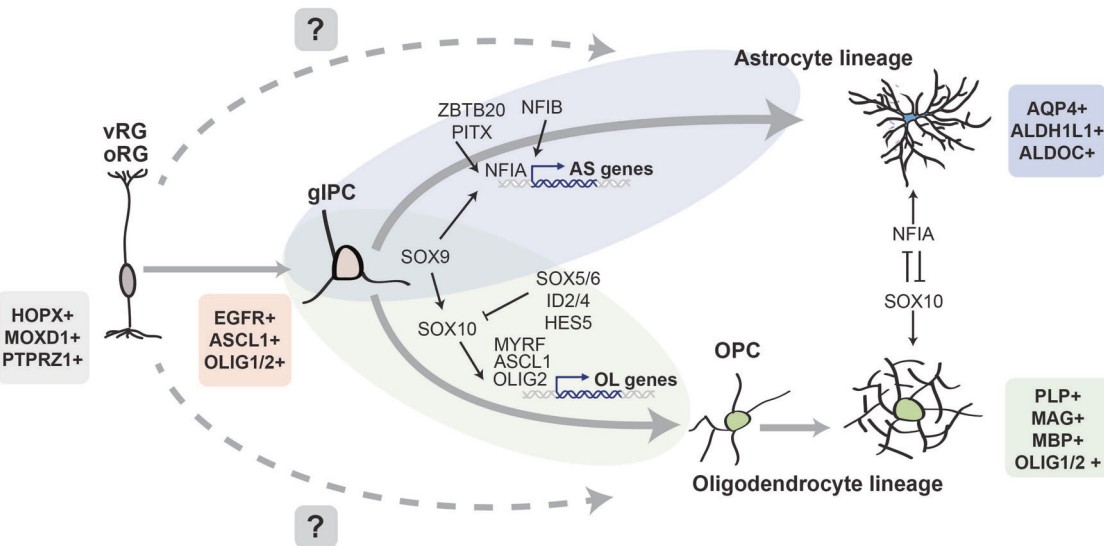

**Fig. 1 Proposed glial differentiation trajectories.** Schematized representation of proposed glial differentiation trajectories. RG are hypothesized to either (solid arrows) give rise to a bipotent glial intermediate progenitor that can generate both astrocytes and oligodendrocytes or (dashed arrows) directly generate astrocyte and oligodendrocyte lineages. Colored boxes indicate lineage markers and relevant TF drivers and inhibitors are listed next to respective lineage types. Outer radial glia (oRG), ventricular radial glia (vRG), glial intermediate progenitor cell (gIPC).

Similar to astrocytes, the OL lineage also demonstrates morphological, transcriptomic, and functional changes throughout maturation. Structurally, OPCs closely resemble NPCs, with a bipolar morphology and a small number of processes that emanate from opposing regions of the soma[76,77]. As OPCs differentiate into postmitotic pre-OLs and pre-myelinating OLs, they expand their total surface area by engaging with neighboring axons, losing their bipolarity, and acquiring filamentous myelin outgrowths[77,78]. This change in morphology coincides with a cascade of TFs binding to regulatory sites of myelination-promoting genes[79]. For instance, during early differentiation, Olig2 is recruited to Sox10 and myelin regulatory factor (Myrf) enhancers, activating their expression[80–82]. The activation of Myrf and additional TFs, including Nkx2-2, Olig1, Ascl1, YY1, Zfhx1b, and Sox10, is necessary for proper OL differentiation into mature myelinating cells[78]. Later in OL development, Olig2 and Brg1 are recruited to the enhancers of cell morphogenesis regulators, such as Cdc42 and Rac1, guiding cytoskeleton reorganization, an important step in the progression toward myelinating OLs[83]. During this shift from pre-myelinating OLs to mature myelinating OLs, myelin structural proteins, including proteolipid protein (PLP), myelin-associated glycoprotein (MAG), and myelin basic protein (MBP), are upregulated, coinciding with increased myelin ensheathment of axons[84,85].

**Tipping the scales: making astrocytes vs oligodendrocytes.** Greater access to primary human tissue specimens has revealed new and diverse progenitor populations, including some that are uniquely hominid[86–88]. Recently, several single-cell RNA-seq papers suggested the presence of a bipotent glial progenitor in the developing human brain that is EGFR+/OLIG2+/OLIG1+/ASCL1+[89–91]. In both the cortex and spinal cord EGFR positive cells are split into two groups—those enriched for astrocyte markers (SOX9 and AQP4) and a separate population expressing canonical oligodendrocyte markers (SOX10, PDGFRA, and PCDH15). These data suggest that at some point, this bipotent glial precursor may diverge towards either an astrocyte or oligodendrocyte trajectory[89,91] (Fig. 1). Notably, there also appears

to be a population of EGFR-negative multipotent intermediate progenitor cells (mIPC), which are enriched for both neuronal and radial glial markers (RBFOX1, ADGRV1, and NRG). Thus, EGFR may serve as a marker for progenitors committed specifically to the glial fate[89].

Assuming astrocytes and OPCs emanate from a shared progenitor, it would be critical that molecular regulators are positioned at the right time and place to assure appropriate proportions and developmental timing of each glial lineage. Transcription factors are one such class of lineage fate determinants that can simultaneously promote one lineage trajectory and repress another. This is evident during gliogenesis when there is strong overlap in the molecular programs that drive AS and OL lineages; however, these shared drivers of development behave in unique and in some cases opposing ways to promote one cellular fate over another.

This concept is perhaps most evident when evaluating the role of SOX9 and its binding partners in determining glial fate specification. SOX9 appears to be an important component of both astrocyte and oligodendrocyte development[25,26,33,92] as Sox9 knockout in the developing spinal cord inhibits both astrogenesis and oligogenesis[25]. However, it serves contrasting roles in each lineage because of differences in when, where, and with which partners it binds. Studies in the developing rodent spinal cord indicate that glial genes are prebound by Sox3 in NSCs. During the initial wave of astrogenesis, genomic sites marked by Sox3 are targeted by Sox9, specifically at regions enriched for Nfi binding motifs[93]. Together, Sox9 and Nfi facilitate the transcription of astrocyte genes to drive early astrogenesis[26] (Fig. 1). In oligogenesis, Sox9 is prebound at multiple oligodendrocyte genes, which are then targeted by Sox10 to facilitate oligodendrocyte development[25,93] (Fig. 1). Unlike in astrocyte development, Sox9 expression appears to peak during the OPC lineage commitment phase of OL development but then drops off during later stages of maturation[94], suggesting that it serves different roles in astrocyte and oligodendrocyte developmental progression.

Not only does SOX9 display differential binding and functional properties in AS and OL lineages, but there is also evidence that the binding partners of SOX9 in one lineage may directly

antagonize SOX9 binding partners of a diverging lineage. Work by Glasgow et al. in chick and mouse models demonstrates that NFIA and Sox10 exhibit antagonizing effects on each other. Expression of SOX10 impedes NFIA-induced expression of AS genes and reciprocally, NFIA inhibits SOX10-induction of OL genes[95]. The same study provided evidence suggesting that Olig2 may play a key role in the NFIA/SOX10 dynamic by reinforcing the interaction between SOX10 and NFIA, promoting a lineage-fate-decision stage[95]. This suggests that while NFIA and SOX10 promote their respective lineages by interacting with SOX9, they also suppress competing lineages by interfering with each other's ability to transcribe specific glial gene sets, thereby tipping the scales toward a specific glial lineage (Fig. 1).

The prospect of a shared AS/OL precursor cell and a precarious scale of AS/OL fate has important implications for glioma research, where malignant cells resemble AS- and OL- like cell types and in some cases, have the capacity to differentiate between the two glial fates.

**Gliomas echo glial development**. Cancer echoes many early developmental principles, including rapid cell proliferation, the activation of nascent developmental signaling pathways, a high degree of cellular plasticity, and susceptibility to local environmental cues. Brain tumors in particular are a prime example of this developmental mimicry. Advancements in single-cell sequencing datasets confirm that brain tumors, especially glioblastomas, exhibit cellular heterogeneity comprised of hierarchies reflective of early neurodevelopment (Fig. 2). This mirroring of early glial lineages may be explained by aberrant activation of developmental regulatory programs, including key TFs, a frequent phenomenon in gliomas[96–101]. Additionally, functional studies in Drosophila and rodent models implicate key fundamental neurodevelopmental signaling cascades, such as Wnt, Notch, and Hedgehog pathways in tumorigenesis[102–106].

Stem-like populations that are abundant in neurodevelopment have also been identified in most primary brain malignancies, including various glioma subtypes, where they are referred to as glioma stem cells (GSCs). While it remains unclear what type of cell(s) these represent and if a pan-GSC marker exists, GSCs exhibit high expression of embryonic stem cell genes and self-renewal capabilities[107–109]. GSCs also demonstrate the ability to self-renew, adapt to the tumor microenvironment, and differentiate into multiple lineage types, reminiscent of the NSC population within the embryonic brain[107–109]. This population is believed to be the source of tumor propagation[108,109] and is capable of evading immune surveillance and therapeutic interventions such as chemotherapy and radiation[110–112]. Essentially, brain tumors recycle early developmental blueprints for generating and maintaining progenitor populations[102–106].

**Glioblastoma**. Glioblastoma (GBM) is classified by the World Health Organization (WHO) as a grade IV glioma. These tumors are the most aggressive and common primary CNS malignancy, accounting for approximately 16% of all primary CNS neoplasms[113]. For primary (de novo) GBMs, which account for 80% of all GBMs, the median age of diagnosis is 62[114]. Secondary GBMs, which develop from lower-grade astrocytomas or oligodendrogliomas, are more frequent in younger adults (mean age 45 years)[114,115]. The typical treatment course for patients with GBM consists of maximal safe surgical resection followed by radiotherapy and temozolomide (TMZ) chemotherapy[116]. Unfortunately, due to the diffuse, heterogeneous, and resilient nature of GBM, these tumors are nearly impossible to entirely irradicate and the prognosis remains bleak with a median survival of 15 months[117,118].

The first GBM datasets included in The Cancer Genome Atlas highlighted inter-tumoral transcriptional heterogeneity across tumors, partitioning them into four transcriptional subtypes—proneural, neural, mesenchymal, and classical—where each exhibits unique cell type-specific gene signatures and oncogenic events[119,120]. However, subsequent studies incorporating multi-region sampling across individual GBM tumors demonstrated that many transcriptional subtypes exist within different regions of the same tumor[121]. This finding was confirmed and delineated by a series of GBM single-cell transcriptomic studies, which uniquely afforded the ability to distinguish between neoplastic and non-neoplastic cells in the tumor bulk using predicted copy number variations for each individual cell. Implementing this approach revealed that the neural tumor signature was likely an artifact of non-cancerous neuronal populations[122] and allowed for more nuanced transcriptional classification of neoplastic cells. Neftel et al. and other groups demonstrated that GBM malignant tumor cells align to neurodevelopmental lineages and generally fall into four transcriptional subtypes that reflect- (1) neural-progenitor-like (NPC-like), (2) oligodendrocyte-progenitor-like (OPC-like), (3) astrocyte-like (AC-like), and (4) mesenchymal-like (MES-like) states, where any given tumor possesses varying ratios of cells that exist in all of these states[123–125]. Further, pseudotime analysis suggests that these cells exist along a stemness hierarchy, with a small population of malignant tumor cells that resemble multipotent NSCs at the apex, and the remaining majority of neoplastic cells existing along the four cellular differentiation trajectories[123] (Fig. 2).

It is important to note that these transcriptomic analyses capture cell states at a single moment in time. Functional studies where cells of a specific GSC population have been engrafted into patient-derived xenografts demonstrate that GSC state is anything but stagnant, and that regardless of the cell population used to initiate the xenograft—AC-like, NPC-like, or MES-like—resulting tumors present all three cell states in comparable frequencies[123,126,127]. Cell state fluctuation may result from endogenous tumor microenvironmental (TME) niches, which have been shown to influence tumor cell biology, including the perivascular[128–131], hypoxic[132–135], and invasive edge[136] niches. Additionally, evidence suggests that therapeutic intervention, itself, can induce a shift in GSC state to a phenotype more conducive for evading harsh treatment strategies[137,138]. However, intrinsic molecular landscape also plays a role in determining GSC state. Neftel et al. found that frequencies of each transcriptional state are associated with genetic alterations in CDK4, PDGFRA, EGFR, and NF1 that appear to bias cell identity towards a particular state[123]. Thus, while GSCs and normal developmental cell types share the capacity to respond to environmental cues, it is the combination of oncogenic mutations, genomic instability, and disruption of chromatin regulators that permits GSCs to override normal systems of checks and balances.

Many of the genetic aberrations in GBM occur in genes that play critical roles in normal glial development. Genetically engineered mouse models (GEMMs) and genetic manipulation of primary glial cell populations illustrate how loss-of-function of GBM-associated tumor suppressor genes (TP53, PTEN, NF1) or gain-of-function of oncogenes (EGFR, PDGFR, RAS, AKT) induce dedifferentiation of quiescent glia[139–143], restrict progenitors to an immature state[144,145], and may even promote the inter-conversion between glial types[146]. Likewise, neurodevelopmental TFs, such as ASCL1, POU3F2, SOX2, SALL2, and OLIG2 can act as oncogenes by inappropriately activating developmental programs that push differentiated GBM cells into tumor propagating GSCs[96,98,147–149]. Olig2 has also demonstrated the capacity to dictate GSC subtype, as a loss of Olig2 causes a shift

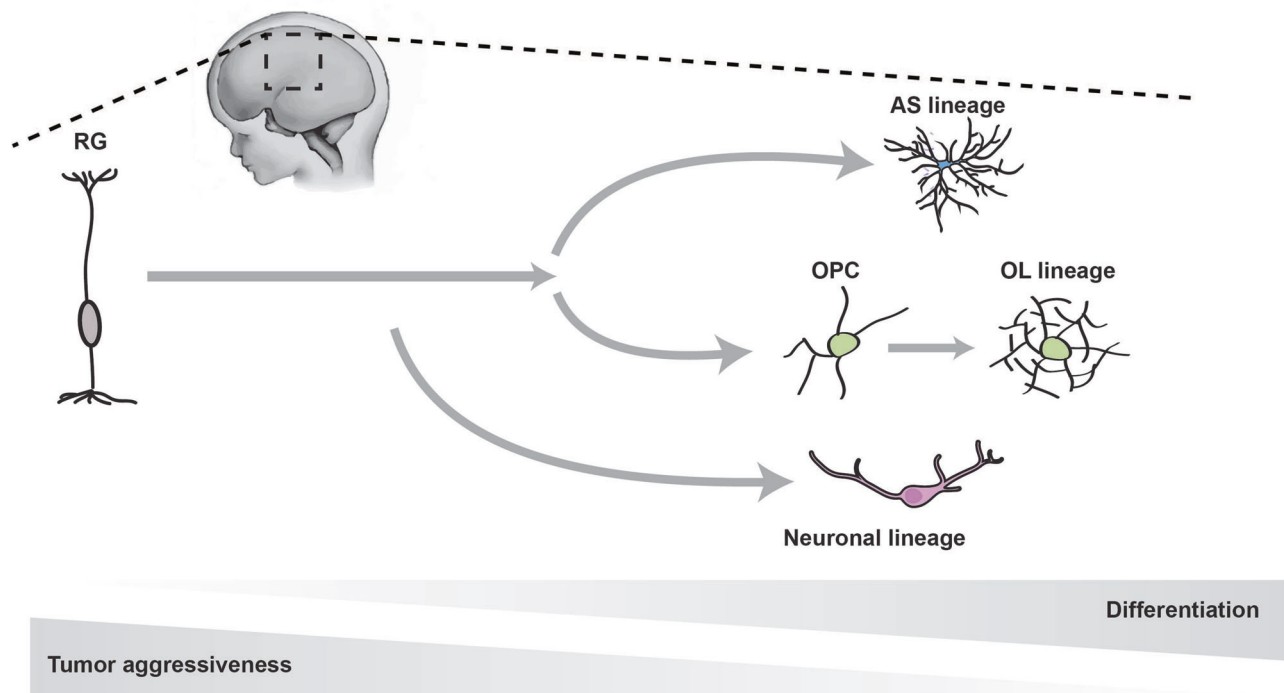

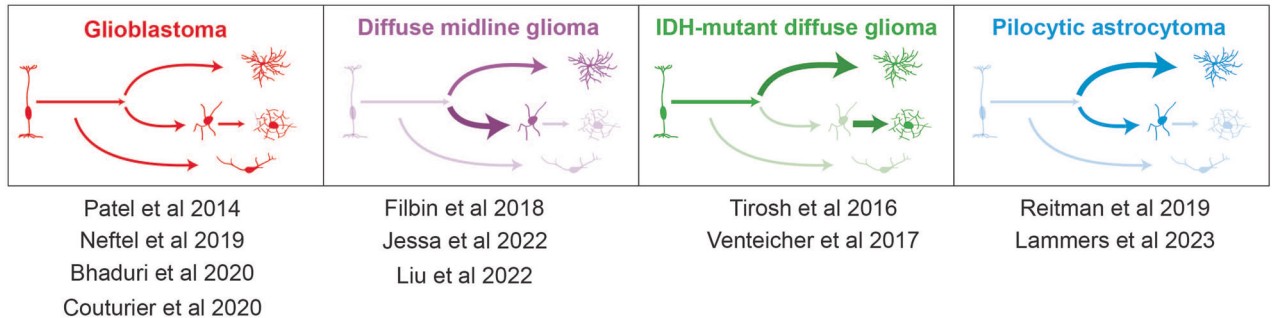

**Fig. 2 Representation of developmental lineages across glioma subtypes.** Schematic depicting the normal neurodevelopmental cell hierarchy and how these cell states are (over)represented across glioma subtypes. The glioma subtypes are arranged in order of relative aggressiveness, with the most aggressive and dedifferentiated (glioblastoma) on the far left. Relative enrichment of neurodevelopmental cell states are represented by arrow thickness. Relevant single-cell publications that support these findings are listed below respective tumor types.

from a proneural transcriptional subtype towards a more astrocytic phenotype, including downregulation of PDGFR and concomitant upregulation of EGFR[150]. Recent studies have also demonstrated reciprocal binding of ASCL1 and OLIG2 in part determines the cell types and degree of migration of tumor cells[151]. When ASCL1 levels are greater than OLIG2, tumors are biased towards astrocyte/NPC-like lineages, whereas the converse scenario pushes cells towards oligodendrocyte fates[151].

Another clear example of the convergence between developmental and oncogenic programs is the redundancy of EGFR activity in development and gliomagenesis. EGFR activity is essential during normal gliogenesis[90,152–154] and both EGFR amplification or constitutively activating mutations (EGFRvIII) are among the most common molecular features of GBM, occurring in about 50% of all cases[155,156]. The tumor biology of EGFR signaling is highly nuanced. Liu et al. demonstrated that the most common EGFR mutation, EGFRvIII, remodels the enhancer regulatory landscape of GBM to induce two key TFs that regulate astrocyte development, SOX9, and FOXG1. Together, these EGFR-dependent TFs work collaboratively to induce oncogenic programs, including c-MYC target genes and EGFR-regulated genes[157]. Interestingly, a recent study from the Deneen group showed that one of these EGFR targets, SOX9, has divergent roles in varying brain tumor subtypes, which each exhibit unique epigenomic states[158]. Thus, while the activation of developmental programs is a shared biological phenomenon across gliomas, individual molecular perturbations can induce opposing outcomes within different cellular contexts.

**Adult-type diffuse gliomas.** Adult-type diffuse gliomas consist of astrocytomas (WHO grades II, III, and IV) and oligodendrogliomas (WHO grades II and III). Unlike GBM, most of the tumors in this class exhibit IDH1/2 mutations, and oligodendrogliomas are further distinguished by the common chromosomal 1p/19 co-deletion[159,160]. IDH-mutant oligodendrogliomas and astrocytomas demonstrate a recycling of early glial differentiation programs to fuel immature developmental cell states. For instance, the regulatory chromatin architecture that is present in normal gliogenesis and the binding of astrocytic TFs like SOX9, NFIA, and BRN2, is shared by models of diffuse glioma

and promoted tumorigenesis[30]. Several of these potent glial fate determinants even demonstrate the capacity to regulate glioma subtype specification reminiscent of the early developmental decision to bias towards AS versus OL lineages. This was perhaps most clearly demonstrated by experiments overexpressing NFIA in a mouse model of oligodendroglioma, which shifted tumor histopathology to more closely reflect astrocytomas[95]. In addition to intrinsic regulators, extrinsic cues also play a role in driving glioma phenotype. PDGF, a potent mitogen involved in generating and maintaining OPCs in the developing brain, induces tumors that reflect oligodendroglioma biology[161,162].

Although diffuse astrocytomas and oligodendrogliomas are characterized by unique histological features, genomic perturbations, and markers of gliogenic regulation, neurodevelopmental lineages are reflected quite consistently between the two tumor types. Single-cell transcriptomic work by Venteicher et al. highlights the similarities between the two glioma subtypes, demonstrating that both harbor three main groups of malignant tumor cells—a relatively small proliferative NSC-like population, and two populations of nonproliferating cells that resemble AS and OL lineages[163] (Fig. 2). Interestingly, the primary differences between astrocytomas and oligodendrogliomas are related to genetic events and tumor microenvironmental niches[163]. When focusing on the cellular heterogeneity within oligodendrogliomas, Tirosh et al. found that CNV-subclones within these tumors span all three transcriptional states—NSC-like, OL-like, and AS-like—suggesting that factors beyond genetic events contribute to the observed developmental hierarchy[164]. This finding is supported by experiments where PDGF exposure yields an inconsistent tumor phenotype between WHO grade II oligodendrogliomas and a mixed oligoastrocytoma profile that expresses both GFAP and Vimentin[161,162].

**Pediatric-type diffuse midline gliomas**. H3 K27-altered diffuse midline gliomas (K27M-DMGs) are a primarily pediatric and extremely aggressive glioma subtype with a median survival of about one-year post-diagnosis[165]. These tumors are regionally specific to midline structures occurring in the thalamus, midbrain, cerebellum, or pons; the latter of which are designated as diffuse intrinsic pontine gliomas (DIPG)[166]. A major breakthrough in the tumor biology of DMGs was the finding that many of these neoplasms contain a lysine27-to-methionine (K27M) mutation in histone 3 (H3). In H3K27-altered DMGs, H3K27M suppresses EZH2, the catalytic subunit of polycomb repressive complex 2 (PRC2). Polycomb activity is involved in a variety of epigenetic regulatory processes, including trimethylation of Lys-27 on histone 3 (H3K27me3)[167,168], which leads to genome-wide dysregulation of gene repression and cell differentiation[168,169].

Single-cell transcriptomic profiling of these tumors has uncovered a similar developmental hierarchy in K27M-DMG to other diffuse gliomas; however, there are several noteworthy differences[170]. K27M-DMGs contain a substantially larger pool of undifferentiated cells, consistent with the more aggressive nature of this tumor subtype[170]. Additionally, undifferentiated cells in K27M-DMGs most closely resemble OPC lineages[170], unlike the putative GSCs in IDH-mutant diffuse gliomas that reflect an NSC identity (Fig. 2).

More recent work from Jessa et al. and Liu et al. implemented a barrage of single-cell genomic, epigenomic, and chromatin profiling approaches to dissect region- and age-related developmental signatures in K27-altered DMG. Jessa et al. profiled cells across DMGs that harbor the H3K27M mutation in different histone variants (H3.1 and H3.3) and demonstrated that while K27M-DMGs appear to maintain a developmentally conserved OPC chromatin signature, differences between H3.1 and H3.3 samples point to distinct OPC developmental origins[171]. Specifically, the molecular profiles of H3.1K27M ACVR1-mutant pontine gliomas resemble early ventral NKX6-1+/SHH-dependent brainstem OPCs, whereas the H3.3K27M signature is more closely aligned with later dorsal PAX3+/BMP-dependent progenitors[171]. These results are supported by work from Michelle Monje and collaborators, which showed that H3.3K27M and H3.1K27M DIPG demonstrate variant-specific PRC2 regulation of developmental gene sets and cell signaling programs[172]. In a similar study, Liu et al. observed the presence of a stem-like OPC population across all H3K27M-DMGs, regardless of age or tumor location[173]. Remarkably, the team identified location-specific OPC subpopulations, where pontine tumors were enriched for a more immature pre-OPC-like signature in comparison with thalamic tumor OPC signatures, corroborating the findings that pontine K27M-DMGs may arise from an OPC population of earlier origins[173]. Recent functional studies also suggests that OPC-like tumor cells play a critical role in fueling K27M-DMG tumor survival by synapsing with surrounding neurons, catalyzing neuronal signaling-induced tumor growth[174–176]. Together, these findings suggest that K27-altered DMGs arising in different brain regions may descend from distinct cells of origin, but likely undergo similar developmental pressures that shape a shared OPC-enriched cellular hierarchy.

Liu et al. further leveraged their age- and region-matched K27M-DMG samples to investigate how patient age, brain tumor location, and mutational status contribute to the global cellular makeup of K27M-DMG tumors. This dataset revealed an enrichment of MES-like cells (a state also observed in adult GBM) in K27M-DMGs from adult patients, compared to region-matched K27M-DMGs from pediatric patients[173]. The authors postulate that this difference may be influenced by endogenous developmental shifts in brain myeloid cell composition, whereby older patients display an enrichment of macrophages compared to pediatric patients who exhibit higher proportions of microglia[173]. When comparing K27M-DMG to age- and region-matched IDH-mutant midline gliomas, it appears that the K27M mutation may skew tumor cells toward a glial/OPC-like cell fate[173]. Lastly, Liu et al. assessed the spatial distribution of cell states in K27M-DMGs identifying physical niches of proliferative OPC-like and OC-like cells, accompanied by larger portions of nonproliferative diffusely-dispersed AC-like cells[173]. This discrepancy between the spatial transcriptomic and existing scRNA-seq datasets may be due to a loss of vulnerable cell populations during tissue dissociation and suggests that spatial transcriptomic approaches, which utilize in-tact tissue sections, might help mitigate these biases in future studies.

**Circumscribed pilocytic astrocytomas**. Unlike grade II and III astrocytomas, pilocytic astrocytomas (PAs) are circumscribed astrocytic gliomas, they do not progress to higher-grade tumors, and most commonly arise in the optic pathway, brainstem, and cerebellum[177]. In comparison to higher-grade diffuse gliomas, PAs have less complex genetics, with most exhibiting only a single-driver alteration activating the MAPK pathway[177]. PAs in the cerebellum commonly develop sporadically and display a somatic rearrangement where the BRAF gene kinase domain is fused to the KIAA1549 gene (referred to as KIAA1549:BRAF)[177]. An additional PA subtype is present in children with Neurofibromatosis type 1 (NF1) who typically experience tumors in optic pathways[177].

Akin to other gliomas, the developmental origins of PAs remain largely unknown, although, there is some evidence implicating OPC and astrocyte populations, specifically. For instance, NF1-deficient astrocytes display hyperactive mTOR

signaling and a greater proliferative capacity, a phenotype that is observed in patient NF1 PA tumors[178,179]. Conversely, limited evidence suggests that ectopic expression of the KIAA1549:BRAF fusion protein does increase proliferation of NSCs in vitro and in vivo; however, there is more uncertainty about the cell of origin in KIAA1549:BRAF fusion PAs as a result of limited experimental models[180,181]. Recent single-cell RNA-seq studies have begun unraveling the cellular hierarchies within these tumors, providing new evidence that an OPC-like progenitor population enriched for MAPK signaling may give rise to a much larger group of AC-like cells with diminished MAPK signaling activity[182,183] (Fig. 2). In comparison to GBM and other diffuse astrocytomas/ oligodendrogliomas, the NSC signature is noticeably absent from PA cells, suggesting that PAs may be driven by a more developmentally committed OPC-like cell[182,183] (Fig. 2).

**How do glioma cells move across developmental time?** While early glial cell types and developmental hierarchies are recapitulated in most gliomas, we are still learning how to use normal developmental trajectories to better understand glioma biology and potential therapeutic interventions. More detailed and complete glial maturation atlases can provide insight into the initial, present, and future developmental stages that glioma cells progress through. This can be thought of as (initial) when in developmental time do gliomas *begin* (i.e., cell of origin); (present) when in developmental time glioma cells *reside* during tumor progression; and (future) when in developmental time glioma cells are capable of *moving towards* with intervention (Fig. 3).

**When in developmental time do gliomas begin?** The first of these questions is a long-standing enigma: which cell types have the capacity to give rise to gliomas? An important note here is that there is certainly a difference between cells with gliomagenic capacity and the reality of which cells originate tumors *in* vivo. There are two prevailing theories addressing glioma cellular origin(s). The first is a scenario where a differentiated somatic cell stochastically gains a combination of oncogenic and/or tumor suppressor mutations, through a variety of possible mechanisms including replication errors or DNA damage, transforming quiescent cells into a stem-like state (Fig. 3). Critics of this theory argue that it is unlikely that a mature non-proliferative cell with a limited lifespan could accumulate the perfect combination of mutations to induce such tumorigenic potential. However, recent work by Simpson Ragdale et al. demonstrates that p53 knockout in reactive astrocytes destabilizes astrocyte fate, such that adult astrocytes are capable of dedifferentiating after chronic injury paradigms[140]. Reacquisition of a stem-like program was largely mediated through age-exacerbated inflammation coupled with EGF secretion from periwound astrocytes, which induces mTOR-dependent reacquisition of early neurodevelopmental TF programs, including Sox2, Olig2, and Ascl1 activation[140]. This suggests that p53 mutation may lift a restraint on fate commitment, while chronic inflammation could serve as a second hit to induce dedifferentiation at later time points.

The second cell-of-origin theory proposes that GSCs arise when an endogenous quiescent stem cell in the brain acquires oncogenic mutations (Fig. 3). Importantly, the exact identity of this stem-like cell is still debated and likely varies across glioma subtypes. Given the cellular heterogeneity of GBM, many hypothesize that the GSC origin in GBM is adult NSCs, which developmentally have the capacity to generate each of the transcriptomic subtypes documented by Neftel et al. (AS-like, OPC-like, and NPC-like). Indeed, a multitude of evidence supports this theory. For instance, GBMs are thought to arise

in the SVZ, where quiescent NSCs reside. Additionally, GBM tumor cells share many properties with NSCs, including high expression of NSC markers[184,185]; they can form neurospheres that have a similar structure to those derived from adult human subventricular zone cells[186,187]; and Nestin-positive tumor cells are critical for tumor growth and chemotherapy resistance[110]. Alternatively, others hypothesize that a lineage-committed precursor, such as an OPC or astrocyte precursor cell, is a more likely culprit. This is because GSCs also express abundant markers for these cell types[188–191] and there is evidence that both lineages possess tumor propagating potential[141,188,192,193]. OPCs vastly outnumber NSCs, which are restricted to ventricular zone niches and the dentate gyrus[194,195]. Additionally, while there is some evidence of adult neurogenesis in the rodent hippocampus, this phenomenon is more controversial in the adult human brain, making OPCs the major proliferative cell population in the adult human CNS[176]. Far less evidence exists on whether astrocyte progenitor cells exist in the adult brain and if so, exhibit the same proliferative potential as OPCs in the adult CNS[196]. It is also conceivable, and highly likely, that there are discrete cells of origin for different GSC subtypes or for the same GSC subtype across different brain regions. The former of these ideas is strongly supported by work from Parada and colleagues, which has demonstrated that the same genetic drivers in different adult lineage-committed progenitors give rise to molecularly and phenotypically distinct GBM subtypes[197].

**When in developmental time do glioma cells reside and thrive during tumor progression?** While it is evident that gliomas reflect multiple early developmental cell types, it is unclear if there are specific maturation stages of glial development reflected in glioma tumors (Fig. 3). This is largely due to our fragmented understanding of glial maturation, which in humans primarily occurs between the third trimester of gestation and the first postnatal month, a brief but critical period when access to primary human tissue samples is greatly restricted. Much of what we do know about glial development and maturation is derived from murine model systems and a limited number of second trimester primary fetal tissue samples. While informative, there are major temporal gaps during this developmental window in human samples, which exhibit many neurodevelopmental differences from rodents[64,198–201]. Curating more comprehensive developmental timelines of glial lineages will help inform whether glioma cells are stalled at particular developmental stages and which molecular programs could be leveraged to coerce maturation towards a quiescent state.

One viable option for building comprehensive timelines of human glial maturation is to leverage human in vitro model systems, such as human brain organoids. There are now numerous robust protocols for forming and culturing human stem cell-induced 3D organoids that are patterned to reflect various regions of the CNS, including forebrain[202,203], midbrain[204,205], hindbrain[206,207], and spinal cord[208,209]. This platform recapitulates many key features of human neurodevelopment including complex cellular composition, intricate tissue architecture, and functionally active neurons[203–205,210–214]. Additionally, long-term culture of human brain organoids depicts maturing astrocyte[215,216] and oligodendrocyte[217–219] lineages with transcriptomic profiles that reflect pre- and postnatal stages of human brain development. Altogether, this makes organoids an ideal system for chronicling elusive windows of development at a high temporal resolution to capture all phases of glial maturation[220].

The human brain organoid model is also well-suited to investigate glioma-glia and glioma-neuron interactions. There are

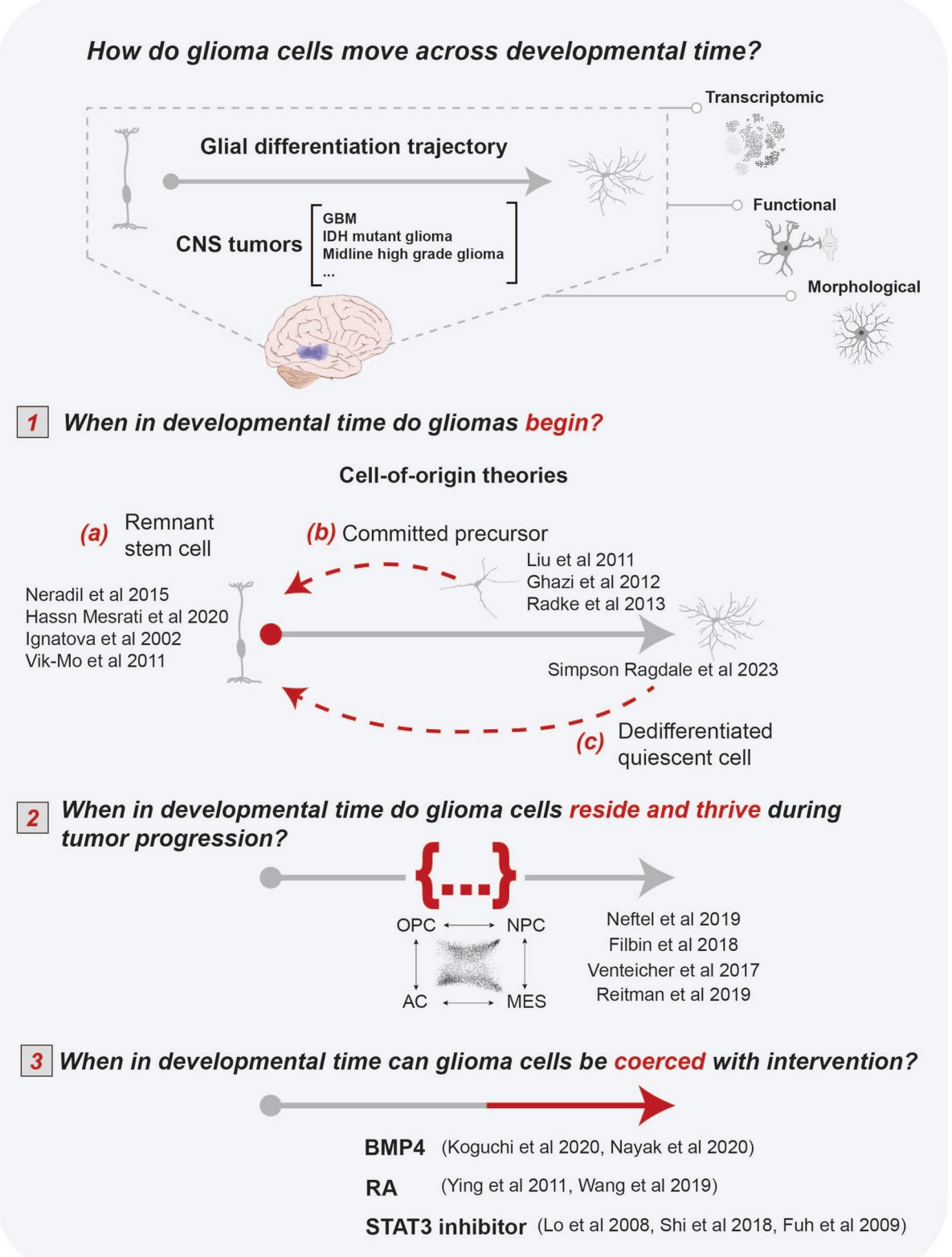

**Fig. 3 How do glioma cells move across developmental time?** Schematized summary of how normal glial differentiation trajectories (generated from transcriptomic, functional, and morphological information) can inform how glioma cells progress through developmental time—when in developmental time do tumor cells (1) begin, (2) thrive during tumorigenesis, and (3) are capable of being coerced to? Citations are listed for primary evidence related to each respective question.

several organoid-based approaches that inherently incorporate both malignant and normal differentiated cell types. These include neoplastic cerebral organoids (neoCOR), whereby specific genetic alterations are introduced at the stem cell stage prior to 3D formation to induce spontaneous tumor formation[221,222], and organoid-glioma coculture models, in which patient-derived tumor cells are engrafted into healthy organoids[127,223–226]. While the neoCOR approach serves as a reductionist system ideal for questions that pertain to specific oncogenic molecular events, the organoid-glioma coculture approach is well-suited for those who wish to account for the diverse and complex molecular nature of human gliomas. Using these models, several groups have demonstrated that GSCs readily invade organoids without losing their proliferative capacity[222–224], form tumor microtubes with host organoid cells[223,225], and maintain a transcriptional profile that closely reflects the parent tumor[127,226]. Given the amenability of organoids to a variety of functional and high-throughput sequencing assays, these models would be optimal for investigating the relationships between malignant tumor cells and the surrounding host tissue.

**When in developmental time are glioma cells capable of moving towards with intervention?** Given the parallels between neurodevelopment and glioma biology, one might reasonably hypothesize that malignant glioma cells are susceptible to the same maturation cues that coerce quiescence in normal glial development (Fig. 3). This is the rationale behind differentiation therapy, which explores therapeutic options to coerce tumor cells through developmental time by exploiting extrinsic and intrinsic factors that regulate cell differentiation and maturation. Of course, a major challenge remains in identifying the optimal glial maturation cues to target in gliomas.

One avenue under active investigation is targeting signaling pathways that are critical in initiating gliogenesis (BMP, Wnt, Notch, STAT3, MAPK/ERK, and TGF-B) and that are frequently hijacked in glioma progression[227]. Several of these pathways appear to have particularly potent impacts on GSC growth, proliferation, and differentiation when targeted through BMP4 and RA treatment[228–231], both of which are important in early astrogenesis[4] and are currently being tested in clinical trials[232]. Another approach is to inhibit glial development TFs that are inappropriately activated in glioma[97,232–234]. For example, STAT3, a key player in the gliogenic switch, is highly upregulated in gliomas, is associated with glioma EGFR amplification, and contributes to GSC proliferation and migration, thus making it a high-priority target for inhibition[233,235]. Multiple groups have identified approaches for suppressing STAT3 activity in GSCs, resulting in increased GSC sensitivity to subsequent chemo and radiation therapy[235–237].

While there is accumulating evidence that malignant glioma cells are receptive to glial developmental cues, there are several technical challenges and caveats to differentiation therapies that must be considered. As illustrated through rigorous testing of BMP4 treatment, it is extremely challenging to identify targetable molecular programs that overcome inter- and intra- tumoral heterogeneity boundaries[238,239]. Not only do tumor cells from different patients exhibit varying genetic backgrounds that respond inconsistently to BMP4 treatment, but there is evidence that GSCs exposed to separate TME niches may also respond uniquely to treatment[240]. Another obstacle is defining the benchmarks for successful GSC differentiation. As demonstrated by in vitro experiments overexpressing TFs to induce normal glial development[31], different TFs will likely induce unique epigenetic and transcriptomic changes. Deciphering which set(s) of changes indicate sufficient maturation will be especially challenging

without more detailed molecular maps of normal maturation in glial lineages. Lastly, even if GSCs respond to differentiation cues to progress through developmental time, these changes may be transient. In fact, Caren et al. demonstrated that GSCs are capable of reverting to a stem-like state following BMP-treatment as a result of incomplete chromatin accessibility changes that permit aberrant SOX TF binding[241]. This suggests that the most effective differentiation method will need to induce large-scale chromatin architecture shifts that are comparable to what occurs in normal glial maturation[234].

## Conclusion

Neurodevelopmental hierarchies are reflected across glioma subtypes, where the programs that drive normal lineage specification and maturation are aberrantly activated to promote tumorigenesis. Utilizing blueprints of normal glial trajectories has revealed new information about glioma occurrence, growth, and resilience. However, there are discontinuities in our current timelines of glial development, particularly between the third trimester and into the first few postnatal months, a critical window of time when immature precursor populations give rise to mature quiescent counterparts. Rapidly evolving model systems, lineage tracing methods, and sequencing platforms will be fundamental for filling in these gaps, and understanding how these elusive developmental stages are represented across gliomas and contribute to tumor progression.

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

## Acknowledgements

We would like to thank Jimena Andersen for helpful discussions. The authors apologize to colleagues whose relevant studies were not cited due to limited space. The research in the author's laboratory was supported by grants from the National Institutes of Health (NIMH R01 MH125956 and NINDS R01 NS123562).

## Author contributions

C.S. composed the manuscript and C.S. and S.A.S. edited and finalized the manuscript.

## Competing interests

The authors declare no competing interests.
