## [Peer Review File · Communications Biology]

Reviewers' comments:

Reviewer #1 (Remarks to the Author):

The review article from Sloan et al. is a well-written and timely piece. The manuscript focuses on the interesting comparison of the time course of astrocyte/oligodendrocyte development and gliomagenesis and provided much needed insight into the parallel and intersection of the two processes. The coverage of relevant literature is thorough, the summarization of research findings is accurately and insightful, the organization and writing of the manuscript is very clear. This article is an excellent synthesis of current understanding in the field and provides thoughtful considerations for areas of investigation in the future. This reviewer supports the publication of this manuscript after minor revision because this article will be an excellent contribution to the field of glioma biology.

Suggested minor revision:

Line 101: "NFIA is not only necessary and sufficient to induce astrocyte formation". While the sufficiency of NFIA to induce astrocytogenesis has been compellingly demonstrated with iPSCs, the necessity of NFIA in astrocyte formation lacks strong support. In Deneen et al., Neuron 2006, the authors found a reduction in the astrocyte marker GFAP in NFIA knockout mice, however, the authors noted that "This reduction in GFAP expression is not due to cell death or cell loss, because there was no change in the total number of cells or in the number of astrocytes, as determined by expression of the complementary NFI protein." Reference 21 and 22 did not contain loss of function study to demonstrate the loss of astrocytes in NFIA deficient mice either. The statement in line 101 should be modified to be more accurate.

Reviewer #2 (Remarks to the Author):

The review "Gliomas: a reflection of temporal gliogenic principles" by Sojka et al is a beautifully written, comprehensive and useful review of the state of the art in normal and oncogenic gliogenesis. I only have a few points to improve the manuscript even more:

- The transition from normal development to gliomas could be slightly more expanded.
- regarding the classification of gliomas, I would recommend using the WHO 2021 classification for all terms. I would also keep pediatric and adult gliomas strictly apart as otherwise readers might be confused. Especially would not mix pilocytic astrocytomas and adult gliomas in and around line 372, as they are not comparable (different drivers, prognosis etc)
In line 439, terminology should be DMG, K27-altered etc.
- In the last 2 weeks several papers on single cell states in normal human (early) embryonal development have been published. I know this was after submission, but would be worth to include in the part that talks about 'what is missing in current knowledge'.
- The part about TFs in gliogenesis should be including OLIG2 and its role in GBM and GSCs.
- In lines 458-459, the authors make the statement that "...DMGs exhibit minimal signatures of differentiated OL-like cells, and only a small percentage of differentiated AC-like cells...". This statement refers to the findings in the original scRNA-seq study of primary DMG reported by Filbin et al 2018. However, a recent study by Liu et al 2022 (which the authors do reference later in the following paragraph in a different context; doi.org/10.1038/s41588-022-01236-3) spatial transcriptomic analysis of primary DMG tissue revealed that AC-like cells surprisingly constitute the

majority of the malignant cell compartment, in contrast to the predominance of OPC-like cancer cells observed by scRNA-seq. The authors could revise the text to more accurately describe our current understanding of the spatial architecture of DMG.

- Space-permitting, the authors could also discuss how OPC-like cells in DMG (and other -like glioma cells) also functionally (not just transcriptionally) resemble normal progenitor cells by engaging in synaptic communication with normal neurons and integrating into neural circuits (doi.org/10.1038/s41586-019-1563-y). A short commentary on how neuronal activity is now known to be crucial regulator of glioma progression would be fitting and make this review more contemporary.

- In line 538-539, the authors should add a note discussing how it is not clear and highly debated whether hippocampal neurogenesis (in the dentate gyrus) occurs in the adult human brain, unlike rodents ([doi/10.1126/science.abn8861](https://doi.org/10.1126/science.abn8861)).

- In lines 542-543 the authors postulate the hypothesis that "... there are discrete cells of origin for different GSC subtypes or for the same GSC subtype across different brain regions". This was actually suggested by a series of studies conducted by the group of Luis Parada that demonstrated that the same genetic drivers in different cells of origin generate molecularly distinct GBM subtype, using genetically engineered mouse models (doi.org/10.1016/j.ccell.2015.09.007). The authors may want to provide a brief review of this important work to strengthen the discussion.

- The authors could comment on how organoids also be employed to study glioma-glia interactions, rather than human glial development alone.

Reviewer 1

The review article from Sloan et al. is a well-written and timely piece. The manuscript focuses on the interesting comparison of the time course of astrocyte/oligodendrocyte development and gliomagenesis and provided much needed insight into the parallel and intersection of the two processes. The coverage of relevant literature is thorough, the summarization of research findings is accurately and insightful, the organization and writing of the manuscript is very clear. This article is an excellent synthesis of current understanding in the field and provides thoughtful considerations for areas of investigation in the future. This reviewer supports the publication of this manuscript after minor revision because this article will be an excellent contribution to the field of glioma biology.

We thank the reviewer for the kind words and hope it will be a great resource for the community.

Line 101: "NFIA is not only necessary and sufficient to induce astrocyte formation". While the sufficiency of NFIA to induce astrocytogenesis has been compellingly demonstrated with iPSCs, the necessity of NFIA in astrocyte formation lacks strong support. In Deneen et al., Neuron 2006, the authors found a reduction in the astrocyte marker GFAP in NFIA knockout mice, however, the authors noted that "This reduction in GFAP expression is not due to cell death or cell loss, because there was no change in the total number of cells or in the number of astrocytes, as determined by expression of the complementary NFI protein." Reference 21 and 22 did not contain loss of function study to demonstrate the loss of astrocytes in NFIA deficient mice either. The statement in line 101 should be modified to be more accurate.

This is a salient point, and the line has been updated to more accurately reflect these findings.

Reviewer 2

The review "Gliomas: a reflection of temporal gliogenic principles" by Sojka et al is a beautifully written, comprehensive and useful review of the state of the art in normal and oncogenic gliogenesis. I only have a few points to improve the manuscript even more:

We would also like to thank this reviewer for the kind words and helpful feedback. We have made the following edits and think the article is now a bit more comprehensive in scope.

The transition from normal development to gliomas could be slightly more expanded.

We have now revised the section "Gliomas echo glial development" (lines 274-283) so that it serves as a smoother transition into the sections on gliomas.

Regarding the classification of gliomas, I would recommend using the WHO 2021 classification for all terms. I would also keep pediatric and adult gliomas strictly apart as otherwise readers might be

confused. Especially would not mix pilocytic astrocytomas and adult gliomas in and around line 372, as they are not comparable (different drivers, prognosis etc). In line 439, terminology should be DMG, K27-altered etc.

We agree with this point and have adjusted the text accordingly, including the removal of the section where pilocytic astrocytomas and adult gliomas were mentioned together. Additionally, we adjusted section headers to be more clear and specified WHO 2021 tumor classification throughout these sections, where needed.

In the last 2 weeks several papers on single cell states in normal human (early) embryonal development have been published. I know this was after submission, but would be worth to include in the part that talks about 'what is missing in current knowledge'.

We have added the following references, but please let us know if there are others related to human embryonal development that we are missing as we want to make sure that we are comprehensive and include all relevant articles.

<https://doi.org/10.1101/2023.10.05.561097>

PMID: 37192616

The part about TFs in gliogenesis should be including OLIG2 and its role in GBM and GSCs.

This is certainly relevant. We have added several additional citations where we mention the role of OLIG2 in GBM (lines 354-360) to emphasize the importance of this TF in GSC maintenance.

In lines 458-459, the authors make the statement that "...DMGs exhibit minimal signatures of differentiated OL-like cells, and only a small percentage of differentiated AC-like cells...". This statement refers to the findings in the original scRNA-seq study of primary DMG reported by Filbin et al 2018. However, a recent study by Liu et al 2022 (which the authors do reference later in the following paragraph in a different context; doi.org/10.1038/s41588-022-01236-3) spatial transcriptomic analysis of primary DMG tissue revealed that AC-like cells surprisingly constitute the majority of the malignant cell compartment, in contrast to the predominance of OPC-like cancer cells observed by scRNA-seq. The authors could revise the text to more accurately describe our current understanding of the spatial architecture of DMG.

We agree that this is an important point made by Liu et al and have added this information to the text (lines 457-473) to highlight the value of spatial data in understanding these tumors.

Space-permitting, the authors could also discuss how OPC-like cells in DMG (and other -like glioma cells) also functionally (not just transcriptionally) resemble normal progenitor cells by engaging in synaptic communication with normal neurons and integrating into neural circuits (doi.org/10.1038/s41586-019-

1563-y). A short commentary on how neuronal activity is now known to be crucial regulator of glioma progression would be fitting and make this review more contemporary.

We agree that these functional studies are foundational in DMG literature and have addressed this work accordingly in the “Pediatric-type diffuse midline gliomas” section.

In line 538-539, the authors should add a note discussing how it is not clear and highly debated whether hippocampal neurogenesis (in the dentate gyrus) occurs in the adult human brain, unlike rodents (doi/10.1126/science.abn8861).

Very true! We have added this information in lines 557-560

.

In lines 542-543 the authors postulate the hypothesis that “... there are discrete cells of origin for different GSC subtypes or for the same GSC subtype across different brain regions”. This was actually suggested by a series of studies conducted by the group of Luis Parada that demonstrated that the same genetic drivers in different cells of origin generate molecularly distinct GBM subtype, using genetically engineered mouse models (doi.org/10.1016/j.ccell.2015.09.007). The authors may want to provide a brief review of this important work to strengthen the discussion.

We agree with this comment and have updated the text accordingly (lines 564-567).

The authors could comment on how organoids also be employed to study glioma-glia interactions, rather than human glial development alone.

We appreciate this suggestion, as it is an important nexus between the organoid and glioma fields. We have included a brief section to cover how organoids can be used to study how glioma cells interact with their environment.

REVIEWERS' COMMENTS:

Reviewer #2 (Remarks to the Author):

Sloan et al. addressed all my comments which improved an already beautifully-written manuscript that delves into the parallels and intersection of normal and oncogenic gliogenesis. I support publication of this manuscript, which will undoubtedly be an excellent contribution to the field of glioma biology.